# Water Body Extraction from Sentinel-2 Imagery with Deep Convolutional Networks and Pixelwise Category Transplantation

Joshua Billson [ID], MD Samiul Islam [ID], Xinyao Sun *[ID] and Irene Cheng [ID]

Multimedia Research Centre, University of Alberta, Edmonton, AB T6G 2E8, Canada
* Correspondence: xinyao1@ualberta.ca

**Abstract:** A common task in land-cover classification is water body extraction, wherein each pixel in an image is labelled as either water or background. Water body detection is integral to the field of urban hydrology, with applications ranging from early flood warning to water resource management. Although traditional index-based methods such as the Normalized Difference Water Index (NDWI) and the Modified Normalized Difference Water Index (MNDWI) have been used to detect water bodies for decades, deep convolutional neural networks (DCNNs) have recently demonstrated promising results. However, training these networks requires access to large quantities of high-quality and accurately labelled data, which is often lacking in the field of remotely sensed imagery. Another challenge stems from the fact that the category of interest typically occupies only a small portion of an image and is thus grossly underrepresented in the data. We propose a novel approach to data augmentation—pixelwise category transplantation (PCT)—as a potential solution to both of these problems. Experimental results demonstrate PCT's ability to improve performance on a variety of models and datasets, achieving an average improvement of 0.749 mean intersection over union (mIoU). Moreover, PCT enables us to outperform the previous high score achieved on the same dataset without introducing a new model architecture. We also explore the suitability of several state-of-the-art segmentation models and loss functions on the task of water body extraction. Finally, we address the shortcomings of previous works by assessing each model on RGB, NIR, and multispectral features to ascertain the relative advantages of each approach. In particular, we find a significant benefit to the inclusion of multispectral bands, with such methods outperforming visible-spectrum models by an average of 4.193 mIoU.

**Keywords:** Sentinel-2; water body extraction; machine learning; CNN; DCNN; fully convolutional networks (FCNs); remote sensing; semantic segmentation; data augmentation

## 1. Introduction

Water body detection from satellite imagery is essential for urban hydrological studies [1], which provides valuable tools for the management of water resources and addressing environmental issues caused by rapid urbanization [2]. Index-based methods have seen widespread study and adoption following the proliferation of remote sensing data. These approaches consider the innate characteristics of water bodies to assign a label of either water or background to each pixel in an image. The Normalized Difference Water Index (NDWI) [3] is a well-known method for extracting water bodies from Landsat imagery, which exploits the tendency for the Near Infrared (NIR) band to be absorbed by water and reflected by dry land and vegetation. Modified Normalized Difference Water Index (MNDWI) [4] improved upon NDWI by replacing NIR with the middle infrared (MIR) band to suppress built-up land. The pixel region index (PRI) [5] is another modification of NDWI, which exploits the local smoothness around a pixel to prevent misclassification of built-up land as water. Feyisa et al. [6] introduced the Automatic Water Extraction Index

(AWEI) to address difficulties in misclassifying built-up land and shadows as water by increasing the contrast between water and other dark surfaces.

Although index-based approaches can work well on controlled datasets, they often fall short for water body detection in real-world conditions as the threshold applied to distinguish water from nonwater can vary with the scene and geographic location [6]. To address the issues associated with traditional index-based approaches, many researchers have been exploring the application of machine learning on the task of automated water body detection [7–9]. Yuan et al. [10] demonstrated the advantage of such methods, with their proposed MC-WBDN model achieving 74.42 Mean Intersection Over Union (mIoU) compared to NDWI and MNDWI, which achieved a score of 3.28 mIoU and 10.44 mIoU, respectively. Following the introduction of Deep Convolutional Neural Network (DCNN), the Fully Convolutional Network (FCN) [11] was introduced to allow for the segmentation of images of arbitrary size by replacing the final dense layers with additional convolutions. Many such models consist of an encoder, which progressively reduces the dimension of successive feature maps while extracting rich semantic information, and a decoder, which restores the resolution of the original image and reintroduces spatial information to facilitate pixel-level classification. To exploit learned feature representations from unrelated visual data, many FCNs employ a pretrained DCNN network such as ResNet [12] as the encoder.

Although the basic encoder/decoder architecture is fairly consistent across FCNs, several noteworthy extensions have been proposed. U-Net [13] introduced a symmetric U-shaped architecture consisting of separate encoder and decoder blocks with skip connections placed between each encoder/decoder pair to facilitate the mixing of spatially and semantically rich features from the encoder and decoder, respectively. UNet++ [14] extends U-Net by introducing additional convolutional layers between the skip connections to bridge the semantic gap between the encoder and decoder. Recurrent convolutions, such as those employed in R2U-Net [15], better replicate the feedback connections in the visual pathway of the human brain and demonstrate improved performance over U-Net in the task of medical image segmentation. Swin-Unet [16] builds on U-Net by employing a type of vision transformer called a Swin Transformer in both the encoder and decoder. Oktay et al. [17] introduced attention gates to the skip connections in the traditional U-Net architecture to suppress irrelevant regions while highlighting salient features. FPN [18] deviates from the U-Net architecture and uses feature pyramids to extract features at various scales, which can then be used for both object detection and semantic segmentation with slight modification. Developed by Google, DeepLabv3+ [19] has a number of novel features, including an Atrous Spatial Pyramid Pooling (ASPP) module between the encoder and decoder to extract features at multiple scales and employing depthwise separable convolutions to reduce computational overhead.

A number of researchers have applied FCNs to water body extraction in recent years. Li et al. [7] used an FCN to detect water bodies in the Beijing metropolitan area from Very High Resolution (VHR) images taken from the GaoFen-2 satellite, achieving an F1 score of 0.92. Moreover, a modified U-Net architecture was able to accomplish a classification accuracy of 98.98% from VHR images captured by both GaoFen-2 and WorldView-2 satellites [8]. More recently, Zhang et al. [20] used FCNs in conjunction with a novel focal loss to perform water body extraction from Synthetic Aperture Radar (SAR) imagery. To address the problem of class imbalance, they reported their results as the Frequency-Weighted Intersection Over Union (FWIoU), for which they achieved a score of 98.54. However, although DCNNs have demonstrated good performance in general, accurate and efficient extraction from complex imagery with abundant micro-water bodies remains a challenge. To address these issues, Kang et al. [21] proposed the Multi-Scale Context Extractor Network (MSCENet) with a novel context feature extractor (CTE) to obtain multiscale features and generate higher-level semantic feature maps with rich contextual information. With this approach, the authors were able to improve accuracy while suppressing background noise from high-rise buildings, vegetation, and shadows, which allowed them to accomplish an F1 score of 0.95.

The recent success of FCNs in water body detection, owing to their ability to extract rich and distinctive feature representations compared to traditional index-based approaches, has made them the commonly adopted method for both water body extraction and land-cover classification. In this study, we explored the performance of various state-of-the-art FCNs on the task of water body extraction. We also examined the impacts of RGB, NIR, and multispectral bands on each of the models. Additionally, we conducted a comprehensive quantitative and qualitative analysis of several current loss functions. Finally, to address the challenge of class imbalance (scarcity of water) in our dataset, we proposed a novel data augmentation approach—Pixelwise Category Transplantation (PCT). Our contributions are as follows.

1. Although previous works have studied the performance of FCNs in the task of water body extraction, there were insufficient discussions on the impacts of band selection on model performance. To address this, we evaluated several state-of-the-art FCNs and compared their performances on RGB, NIR, and multispectral features.

2. Water body extraction based on machine learning is frequently subject to class imbalance in the dataset. This is often due to the category of interest occupying only a small portion of an image. The traditional solution to this problem is to design the loss function to place greater emphasis on the minority class. Thus, we examined a variety of previously proposed loss functions and provided a thorough analysis of their relative performances on our dataset.

3. While FCNs have already demonstrated promising results in water body detection, the training of such models requires access to large quantities of accurately labelled data. Unfortunately, a sizeable database of appropriately labelled images, such as ImageNet [22], is not presently available for remotely sensed data [23]. Furthermore, water is often underrepresented in many datasets due to occupying only a small portion of an image. To address these issues, we introduced PCT, a novel form of data augmentation applicable to both water body extraction and image segmentation in general.

## 2. Materials and Methods

### 2.1. Data Source and Preparation

2.1.1. Characteristics and Exploratory Analysis

The dataset used in this study was collected by Yuan et al. [10] and consists of Sentinel-2 satellite imagery captured over Chengdu City in Sichuan Province, China. Chengdu City is located on the edge of the Sichuan Basin and features a number of prominent water bodies, including the Jin, Fu, and Sha Rivers (see Figure 1). The surrounding region has a humid subtropical climate and receives over 850 mm of precipitation annually. Each sample is comprised of two 16-bit rasters of size 20,976 × 20,982 (RGB and NIR), a 16-bit raster of size 10,488 × 10,491 (SWIR), and a corresponding ground-truth mask showing the appearance of water. It should be noted that the resolution of the SWIR band is half that of RGB/NIR, and unless otherwise stated, all resolutions from this point on are given with respect to RGB and NIR. Images were created as a composite of scans selected over the course of a month to remove cloud cover, and atmospheric correction was performed in ArcGIS. Separate samples are provided for three different timestamps corresponding to the months of April 2018, December 2018, and February 2019. Each timestamp is subject to different lighting and atmospheric conditions, which presents the opportunity to evaluate our methods under a variety of circumstances. An example of the data from timestamp 1 (April 2018) is provided in Figure 2.

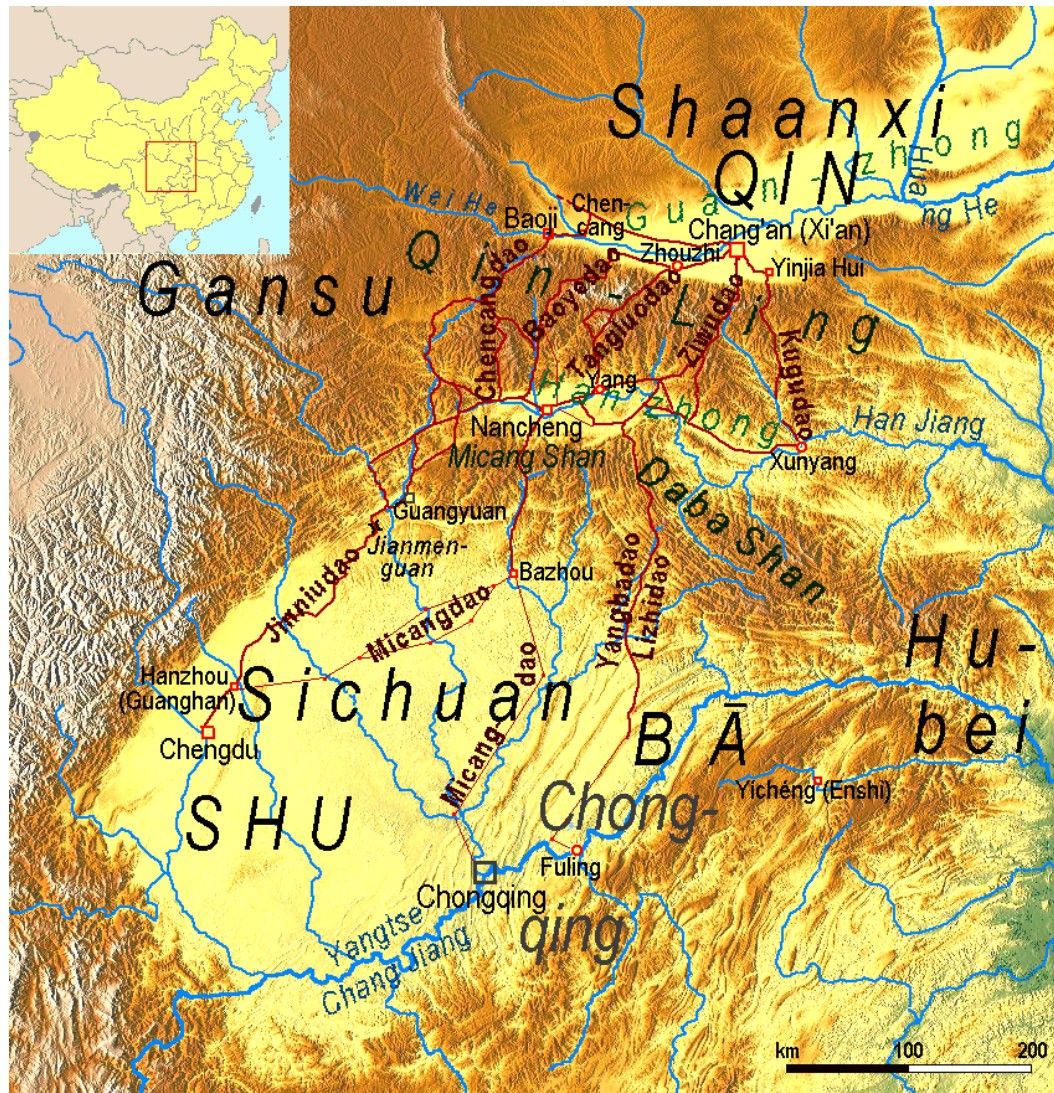

**Figure 1.** Our study area is Chengdu City in Sichuan Province, China. The city is located on the edge of the Sichuan Basin and features a number of prominent water bodies, including the Jin, Fu, and Sha Rivers. Here we provide a map of the Sichuan Basin [24], which documents some significant water bodies in the region.

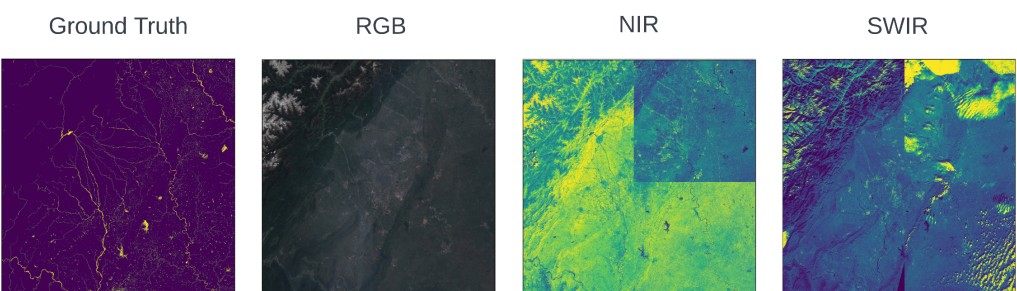

**Figure 2.** The dataset used in this study was collected by Yuan et al. [10] and consists of Sentinel-2 imagery captured over Chengdu City in Sichuan Province, China. Each sample is comprised of two 16-bit rasters of size 20,976 × 20,982 (RGB and NIR), a 16-bit raster of size 10,488 × 10,491 (SWIR), and a corresponding ground-truth mask showing the occurrence of water. Each image was created as a composite of pictures selected over a month to remove cloud cover.

Our exploratory analysis revealed that the dataset is highly imbalanced with respect to water, with only 2.18% of the pixels in the 20,976 × 20,982 ground-truth mask being classified as water. Moreover, the distribution of water content over tiles is highly right-skewed, as demonstrated in Figure 3. In particular, we observed that only 29 of the 400 tiles in our dataset contain at least 5% water, and 41 tiles contain no water pixels whatsoever. To address this problem, we evaluated a variety of loss functions that have previously demonstrated promising results on imbalanced data sets [25–27]. However, while such losses can alleviate many of the problems associated with class imbalance, they cannot accommodate cases in which a class is missing entirely. One solution to this issue is to simply discard all samples for which water is absent. Although such an approach is easy to implement, it is often undesirable to discard data, particularly when the dataset is already limited. In our case, such a solution would throw away 10% of our data. To address this issue, we propose PCT, a novel means of data augmentation which directly addresses the cause of class imbalance by transplanting water features from one tile to another.

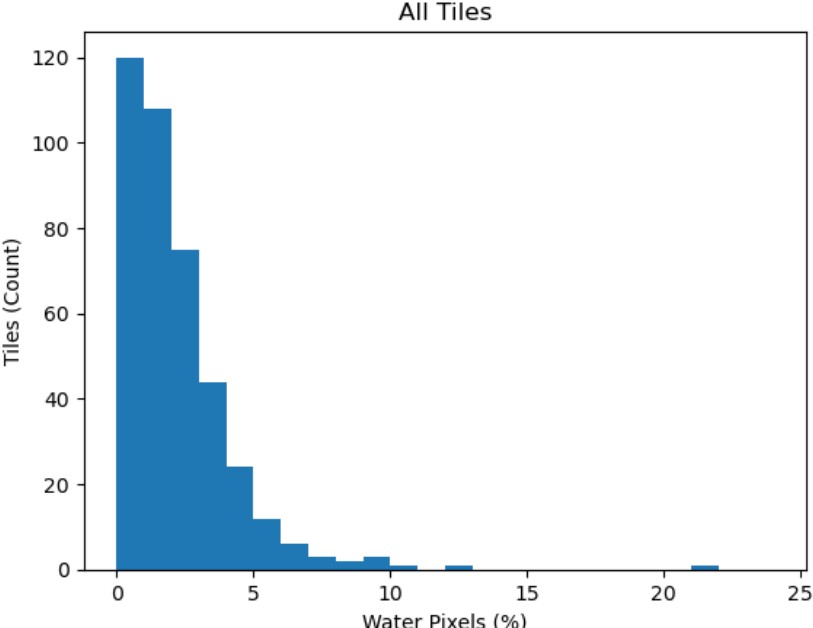

**Figure 3.** The distribution of water over tiles is highly right-skewed. In total, 2.18% of the original 20,982 × 20,982 pixels are classified as water, and only 29 out of 400 tiles contain at least 5% water.

2.1.2. Data Generation Pipeline

Our first step is to cut the image into 400 1024 × 1024 tiles. To address issues regarding memory overhead, each tile is saved to disk and read on demand during training and evaluation. Our models expect an input size of 512 × 512, which necessitates further partitioning of the tiles. We implemented multiple algorithms for this purpose. A nonoverlapping partitioning strategy was used to create both test and validation data (Figure 4a). Moreover, two approaches for generating partially overlapping patches were developed to increase the number of samples available for training. In the first approach, each tile was partitioned into nine half-overlapping patches (Figure 4b). This method was used in our baseline experiments. In the second approach, 512 × 512 patches were randomly sampled from each tile to prevent the model from repeatedly observing water bodies in the same neighbourhood during training (Figure 4c). This proved necessary to minimize overfitting when applying our proposed PCT algorithm for data augmentation. To prevent our model from ever seeing features during testing/validation that it has already observed in training, we never allow for the creation of overlapping tiles nor the generation of patches spanning two or more adjacent tiles.

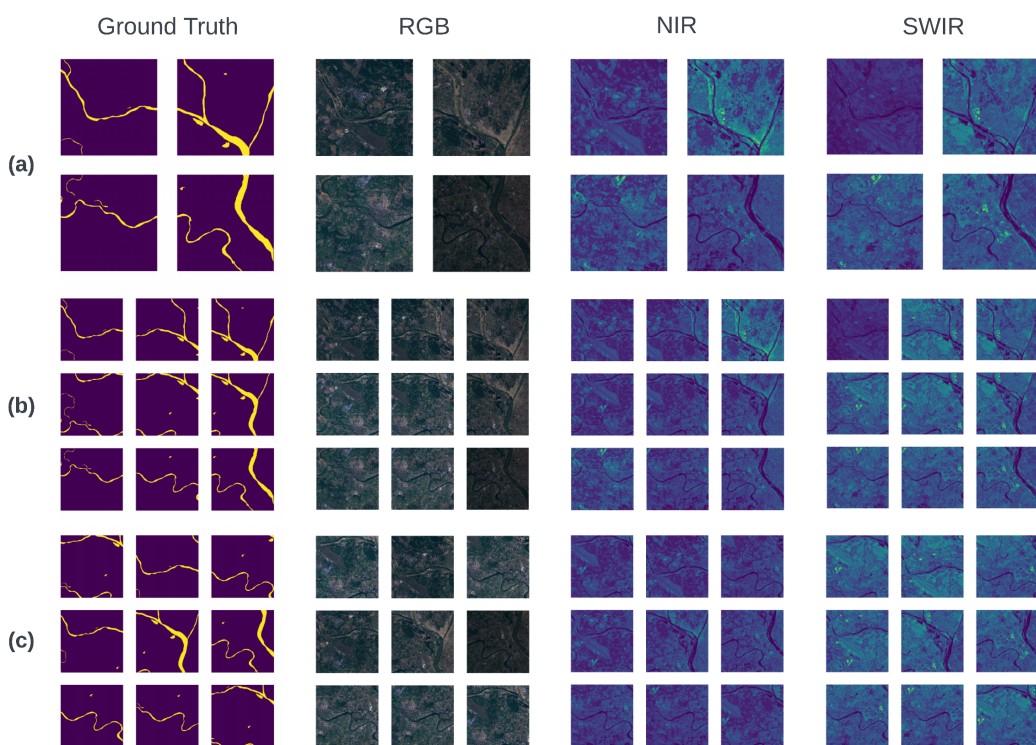

**Figure 4.** A demonstration of several partitioning strategies in which a 1024 × 1024 tile is cut into 512 × 512 patches. (**a**) A nonoverlapping partitioning strategy in which each 1024 × 1024 tile is cut into four 512 × 512 patches. (**b**) An overlapping partitioning strategy where each tile is partitioned into nine half-overlapping patches. (**c**) An overlapping partitioning strategy where 9 512 × 512 patches are randomly sampled from each tile at the start of each epoch.

As previously noted, the resolution of the SWIR band is half that of the RGB and NIR bands. Thus, the first stage in the data generation pipeline is to upsample the SWIR band by a factor of two in each dimension via bilinear interpolation. Next, we clip all pixel intensities to a maximum of 3000 in the NIR and SWIR bands to help filter out any remaining clouds. We observe that the intensities for each channel are approximately Gaussian. Thus, we normalize each channel $X_i$ by subtracting the mean and dividing by the standard deviation, as shown in Equation (1). Following normalization, a series of basic augmentation operations are applied, consisting of a random 90° counter-clockwise rotation with 25% probability and a random horizontal and vertical flip with 50% probability. In later experiments, we will introduce our proposed PCT algorithm for augmenting by water body transplantation.

$$X_i = \frac{X_i - \mu(X_i)}{\sigma(X_i)}. \tag{1}$$

## 2.2. Methodology

### 2.2.1. Loss Function

Although a large number of loss functions have been proposed for the task of semantic image segmentation, of particular interest to us is their performance under class imbalance owing to the severe underrepresentation of water in our dataset. Additionally, our region of interest contains a large number of long and thin water bodies, which are found to be difficult for DCNNs to detect accurately, particularly in built-up locations and under adverse atmospheric and lighting conditions. We thus examine the performance of various popular loss functions on our dataset. For each of the formulas below, we define $\Omega$ as the set of all pixels in the image, $l_i$ as the ground-truth label for the $i^{th}$ pixel, and $p_i$ as the prediction for the $i^{th}$ pixel.

Binary Cross Entropy (BCE) is a well-known loss for binary classification problems. To use this loss in semantic segmentation, we first compute the BCE loss for each pixel, then return the mean loss over all pixels. To address the problem of class imbalance, two weighting factors $w_0$ and $w_1$ may be introduced, where $w_0$ is the weight applied to negative labels (background) and $w_1$ is applied to positive labels (water). Setting $w_0 = w_1 = 1$ gives us the standard BCE loss, which is given by the following formula:

$$loss_{BCE} = -\frac{1}{|\Omega|} \sum_{i=1}^{|\Omega|} l_i \cdot log(p_i) + (1 - l_i) \cdot log(1 - p_i). \tag{2}$$

The Dice and Jaccard losses are widely used in the field of semantic image segmentation and are typically interchangeable. Both of these can be considered region-based losses, which encourage our model to maximize the intersection between the prediction and ground truth while minimizing the difference. Another benefit of both Jaccard and Dice losses is their resilience to class imbalance. The formulas for both are given as

$$loss_{Dice} = 1 - \frac{2 \cdot \sum_{i=1}^{|\Omega|} p_i l_i}{\sum_{i=1}^{|\Omega|} p_i + \sum_{i=1}^{|\Omega|} l_i} \tag{3}$$

$$loss_{Jaccard} = 1 - \frac{\sum_{i=1}^{|\Omega|} p_i l_i}{\sum_{i=1}^{|\Omega|} p_i + \sum_{i=1}^{|\Omega|} l_i - \sum_{i=1}^{|\Omega|} p_i l_i}. \tag{4}$$

The focal loss was proposed by Lin et al. [28] as a solution to the class imbalance problem. The loss function is a dynamically scaled cross-entropy loss where the scaling factor reduces the contribution of confident predictions to focus the model on learning "hard" examples. The assumption is that instances belonging to the underrepresented class typically fall into the set of "hard" samples. This is accomplished by introducing the parameter $\gamma$, which serves to diminish the contribution of confident predictions. In Equation (5), we observe how setting $\gamma > 1$ causes the weighting factor $(1 - q_i)$ to approach 0 as the confidence in our prediction increases to a maximum of 1:

$$loss_F = -\frac{1}{|\Omega|} \sum_{i=1}^{|\Omega|} (1 - q_i)^\gamma log(q_i) \tag{5}$$

$$q_i = \begin{cases} 1 - p_i & l_i = 0 \\ p_i & l_i = 1. \end{cases}$$

The Tversky index ($TI$) is a generalization of the Dice similarity coefficient and F-score. The Tversky loss [26], defined as $1 - TI$, is intended to solve the problem of class imbalance in semantic segmentation by introducing a weighting factor to increase the contribution of false negatives. By emphasizing false negatives, we encourage our model to learn a higher recall with respect to the minority class. We observe in the following formula that setting $\alpha > 0.5$ places greater emphasis on false negatives while diminishing the contribution of false positives:

$$loss_T = 1 - \frac{TP}{TP + \alpha \cdot FN + (1 - \alpha) \cdot FP} \tag{6}$$

$$TP = \sum_{i=1}^{|\Omega|} p_i l_i \quad FN = \sum_{i=1}^{|\Omega|} (1 - p_i) \cdot l_i \quad FP = \sum_{i=1}^{|\Omega|} p_i \cdot (1 - l_i).$$

An extension of the Tversky loss, focal Tversky loss [27] introduces a hyperparameter $\gamma \in [1, 3]$ with the goal of downweighting confident predictions while emphasizing hard examples. The formula for the case of binary classification is defined as

$$loss_{FT} = (loss_T)^{\frac{1}{\gamma}}. \tag{7}$$

Previous researchers [10] have achieved good results by combining a region-based loss, such as Dice or Jaccard, with a pixelwise loss like BCE. While the pixelwise loss seeks to minimize the error averaged over all individual pixels, the region-based loss aims to optimize the smoothness of regions to improve the mIoU. Based on the results of our loss function evaluation (discussed in Section 3), we selected a combination approach for all further experiments, with BCE and Jaccard serving as the pixelwise and region-based losses, respectively. In the formula for our loss, given by Equation (8), we observe that setting $\alpha < 0.5$ places greater emphasis on the regional loss while setting $\alpha > 0.5$ emphasizes the pixelwise loss. Our evaluation demonstrated good performance when emphasizing both components equally, and thus we set $\alpha = 0.5$. Our loss is then defined as

$$loss = \alpha \cdot loss_{BCE} + (1 - \alpha) \cdot loss_{Jaccard}. \tag{8}$$

### 2.2.2. Spectral Contribution in Water Body Detection

Although a number of previous works have studied the application of FCNs in the task of water body extraction [10,21], there has been insufficient research into the effects of spectral band selection in such methods. To address this, we examine the relative performance of RGB-only, NIR-only, and multispectral models across a variety of architectures. We first define a base architecture for each evaluated model. Next, we select an input layer that takes the chosen bands from the data pipeline, concatenates them along the channel dimension, and performs a 2D convolution to produce a tensor with the number of channels expected by the base model. For example, in the case of multispectral U-Net, we concatenate the RGB, NIR, and SWIR bands to produce a tensor with five channels, then perform a convolution to output a tensor with 64 channels which is fed into the base U-Net model.

### 2.2.3. Experimental Procedure

All experiments were implemented in TensorFlow and executed on an RTX 2080 GPU. Each model typically took between 3 to 8 h to train, depending on the architecture and batch size. We divided our dataset of 400 tiles into 250 training samples, 50 validation samples, and 100 test samples. We shuffled the tiles before partitioning in order to ensure a comparable distribution of features across samples. Each model was trained on the entirety of the training data for 50 epochs or until we observed that the validation loss had not improved for 10 consecutive epochs. Apart from the experiments comparing loss function performance, all models were trained with the loss given by Equation (8). The model's performance was evaluated on the validation data after each epoch to track its learning rate and to allow for early stopping. Once training had concluded, the model was evaluated on the test set to provide an estimate of generalized performance. Depending on the memory limitations imposed by each model, we utilized a batch size of either 4, 2, or 1. For optimization, we employed the Adam algorithm [29] with an initial learning rate of $5.0 \times 10^{-5}$, which was halved every 10 epochs. Predictions were saved for qualitative analysis every five epochs during training, and again once training had concluded.

### 2.3. Pixelwise Category Transplantation

Class imbalance is a common problem in the task of image segmentation, as categories of interest typically occupy only a small portion of a field of view. When a category of interest is severely underrepresented in the training data, it is challenging to learn a model that can accurately discriminate instances of that class. An additional issue facing

researchers in the field of remote sensing is the absence of an extensive standardized database of labelled images in the vein of ImageNet [23]. Here we propose a novel data augmentation approach—PCT—to simultaneously address both of these issues.

In essence, PCT involves the transplantation of pixels belonging to some category of interest from a source tile to a destination tile. This may be repeated any number of times, facilitating the creation of augmented tiles possessing an arbitrary representation of the transplanted category. For example, we may repeatedly transplant water bodies from multiple sources onto a single destination until achieving a water content of at least $\theta$%, where $\theta \in [0, 100]$ is defined here as the percentage of pixels belonging to our category of interest (water). In this way, we increase the prevalence of underrepresented classes while simultaneously introducing new features to train our model. Figure 5 provides a visual depiction of our proposed algorithm, which is given by the formula

$$T_{PCT} = \left( M \odot T_{src} \right) + \left( \tilde{M} \odot T_{dst} \right),\tag{9}$$

where $T_{PCT}$, $T_{src}$, and $T_{dst}$ are the augmented tile after applying PCT, the tile from which we want to extract pixels of interest (water), and the tile to which we want to transplant the pixels of interest (water), respectively. We define $M$ as the mask for the source tile $T_{src}$, where all pixels belonging to our category of interest are set to 1 with all others being 0, and $\tilde{M}$ as the result of applying a bitwise NOT operation to $M$. Elementwise multiplication, also known as the Hadamard product, is denoted by the $\odot$ operator. The result of PCT applied to the RGB, NIR, and SWIR bands is shown in Figure 6.

One challenge with this approach is the risk of overfitting owing to the model repeatedly observing the same transplanted features. We combined two techniques to address this issue. The first was to slice our tiles into patches by using a random subsampling strategy as depicted in Figure 4c, which ensures that the model rarely observes the same water body in the same position twice. Our second technique is to repeatedly transplant water features from multiple tiles with low water content rather than transplanting from a single tile with high water content. This approach provides a few benefits, the first of which is that there are many more tiles with low water content than high water content, thus providing us with a greater number of source tiles from which to transplant. Moreover, by combining water features from multiple source tiles, we can create unique combinations of features from diverse water bodies. By taking these two approaches together, we can address the issue of overfitting in our method.

To demonstrate our approach's efficacy under various conditions, we train each of the models outlined in our Baseline Model Evaluation section on several multispectral datasets. For each timestamp in our initial dataset, we have five different subdatasets on which to train: a baseline dataset to which no augmentation has been applied and four augmented datasets where PCT was applied under a $\theta$ value of 5%, 10%, 15%, and 20%. The level of transplantation, dictated by $\theta$, then becomes a hyperparameter which can be tuned to achieve optimal performance. Because we have data at three different timestamps, there are 15 distinct datasets to train. For the sake of expediency, only a single model was trained on all three timestamps, and the remainder were trained exclusively on timestamp 1. For each model, the baseline performance was taken from the results of our experiments in the Various Spectral Contribution In Water Body Detection section.

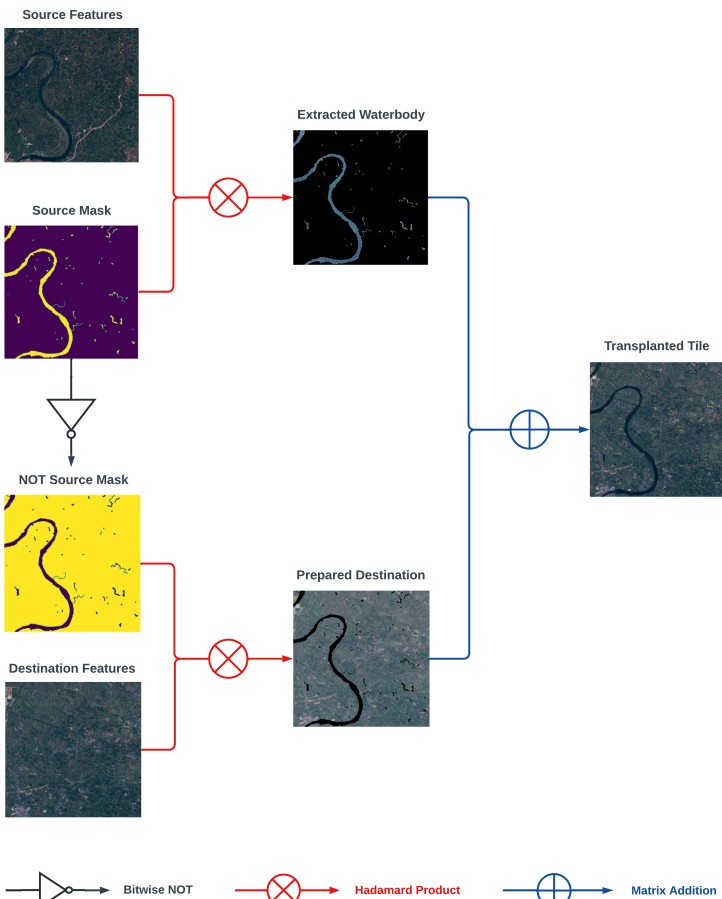

**Figure 5.** The first step of PCT is to perform an elementwise multiplication between the mask and the source feature and between the NOT mask and the destination features in order to extract the water body for transplantation and to create a space in which to place the transplanted water, respectively. The final step is to perform elementwise addition between the two resulting feature maps to produce the final transplanted tile.

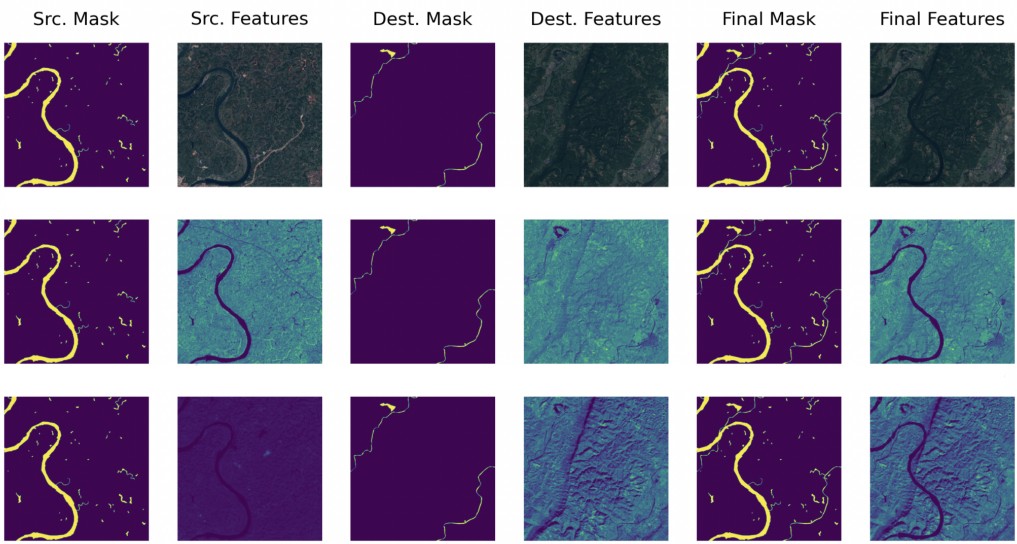

**Figure 6.** A demonstration of PCT with respect to water bodies. Each row depicts a different input band (RGB, NIR, and SWIR, respectively), and the columns show each band's source, destination, and final mask/features. This process can be repeated to transplant multiple water bodies as desired.

## 3. Results

### 3.1. Loss Function Evaluation

Table 1 summarizes the results of our loss function analysis. For each loss, we trained a multispectral R2U-Net model for 50 epochs, after which we recorded the final performance on the test set with respect to mIoU, precision, and recall. It is immediately apparent that a combined loss consisting of BCE and Jaccard achieves the best overall performance while striking a good balance between precision and recall. Based on these results, we chose to make exclusive use of Jaccard + BCE for all future experiments. Further analysis of Table 1 reveals a pattern of precision–recall tradeoff where losses generally achieve high recall at the expense of precision or vice versa. This is supported by Figure 7, which demonstrates a strong negative correlation between the two metrics. In particular, we observe that high-recall, low-precision models are penalized more severely than the reverse by mIoU due to the significant class imbalance present in our dataset. For example, weighted BCE achieved both the highest recall and the lowest precision while also earning the worst mIoU out of all evaluated losses. In contrast, the best performance was seen in models achieving a good balance of both precision and recall, as is the case with those trained on Jaccard + BCE.

**Table 1.** Loss Function Evaluation.

| Loss Function | Performance (mIoU) | Precision (%) | Recall (%) |
| --- | --- | --- | --- |
| Weighted BCE | 62.007 | 30.596 | 92.019 |
| BCE | 73.537 | 77.830 | 57.299 |
| Focal | 73.403 | 76.872 | 57.724 |
| Focal Tversky | 73.900 | 61.504 | 75.802 |
| Tversky | 74.131 | 62.422 | 76.111 |
| Jaccard | 74.509 | 70.218 | 65.538 |
| Dice | 74.943 | 70.809 | 68.818 |
| Dice + BCE | 75.235 | 72.673 | 67.492 |
| Jaccard + BCE | 75.420 | 73.066 | 68.026 |

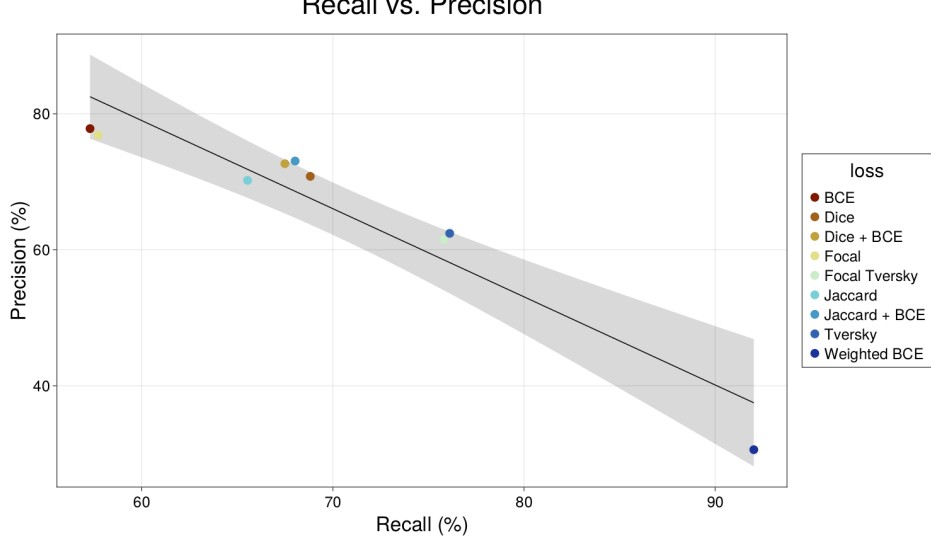

**Figure 7.** A comparison of recall vs. precision for multispectral R2U-Net trained on a variety of losses. We observe a strong negative correlation between the two metrics.

A qualitative comparison of each loss function is provided in Figure 8. We observe that high recall losses such as weighted BCE and Tversky perform the best on the task of extracting small and thin water bodies. Moreover, we observe that weighted BCE, which has the highest recall among all the evaluated losses, is the only one to consistently extract thin and micro-water bodies. However, it also tends to grossly overestimate the water's boundaries, as shown in Figure 8d. Additionally, Figure 8c demonstrates the tendency for high recall functions to misclassify low-albedo surfaces as water. Although the extraction of thin rivers and small water bodies appears challenging for most losses, it seems that high-precision losses struggle more than others, as illustrated in Figure 8a,b. In particular, we observe that BCE, which has the highest precision out of all the evaluated losses, struggles with extracting anything other than large and medium-sized water bodies. As with the quantitative analysis, we observe that the best overall performance is produced by loss functions which strike a good balance between precision and recall.

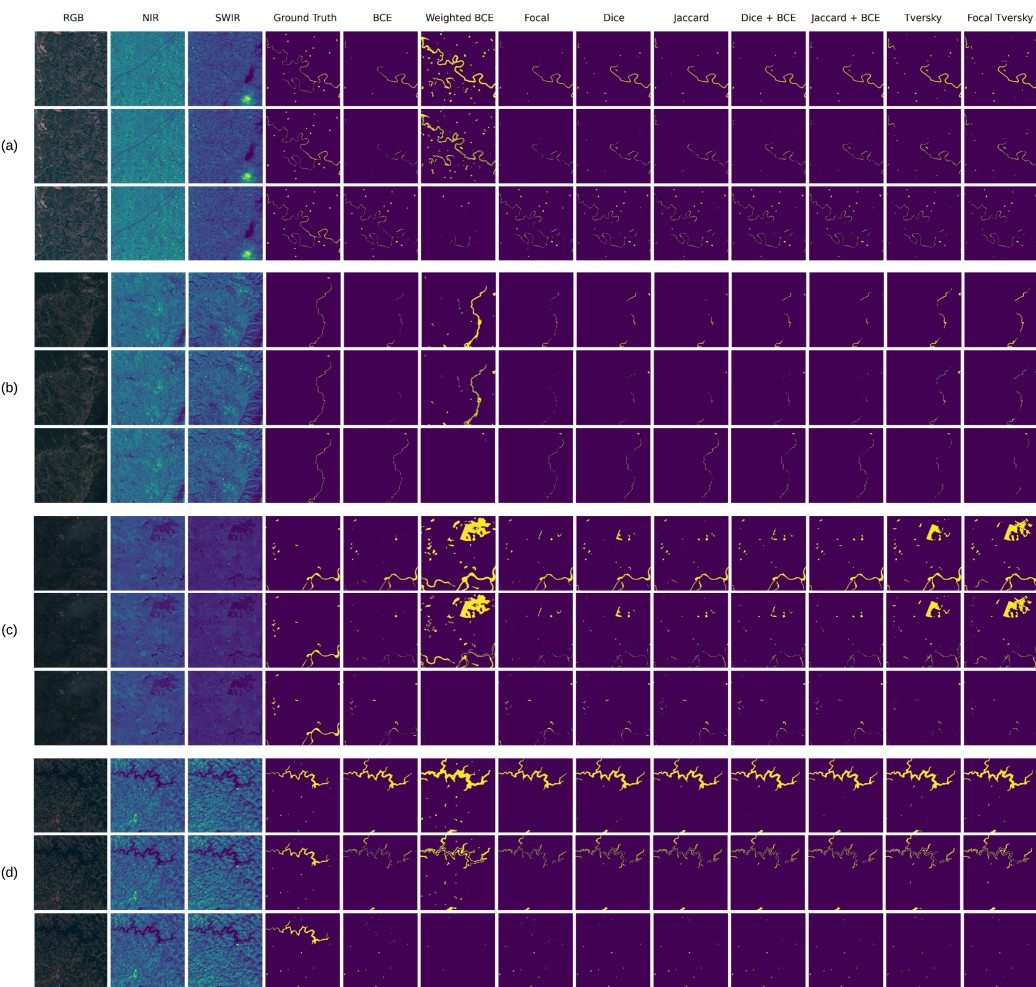

**Figure 8.** Qualitative comparison of multispectral R2U-Net trained with various loss functions. For each example, the first row shows raw predictions, the second row shows false positives, and the third row shows false negatives. (**a**) A thin river is only extracted fully by weighted BCE. (**b**) A thin river proves challenging for most losses. (**c**) The fields in the top right produce false positives in many losses. (**d**) A large contiguous water body is fully extracted by all loss functions.

*3.2. Baseline Model Evaluation*

Table 2 summarizes the test performance for a variety of FCNs on RGB, NIR, and multispectral inputs before applying our proposed PCT algorithm. The performance of NDWI and MNDWI was previously evaluated by Yuan et al. [10] on the same dataset, which is provided here to allow for comparison against deep learning-based approaches. We observed generally good performance among U-Net-inspired architectures, which are characterized by symmetrically arranged encoder and decoder blocks joined by skip connections to combine spatially rich feature maps with semantically rich features from the encoder and decoder, respectively. Of all the evaluated models, R2U-Net performs the best on both multispectral and RGB inputs, achieving the best performance of 75.420 mIoU. In contrast, the worst performance was observed in FPN across all input bands. We chose a ResNet50 [12] encoder for DeepLabV3+, which was evaluated with both randomly initialized and ImageNet pretrained weights. We observed significant gains for the model with the ImageNet pretrained encoder, particularly on RGB features, with an improvement of 7.281 mIoU. Smaller gains were also observed for NIR and multispectral features.

**Table 2.** Baseline Model Evaluation

| Model | RGB (mIoU) | NIR (mIoU) | Multispectral (mIoU) |
|---|---|---|---|
| NDWI [3] | - | - | 3.280 |
| MNDWI [4] | - | - | 10.440 |
| FPN [18] | 63.888 | 69.562 | 71.408 |
| DeepLabV3+ [19] | 65.674 | 71.737 | 72.032 |
| Swin-Unet [16] | 67.726 | 70.675 | 73.264 |
| U-Net [13] | 71.952 | 73.849 | 74.532 |
| U-Net++ [14] | 71.309 | 73.308 | 74.551 |
| Attention U-Net [17] | 71.061 | 72.870 | 74.763 |
| DeepLabV3+ (ImageNet) [19] | 72.955 | 74.791 | 75.403 |
| R2U-Net [15] | 73.267 | 73.570 | 75.420 |

Figure 9 presents a qualitative analysis of different model architectures. We observe that large water bodies are easily extracted by all models, while thin rivers and micro-water bodies present more of a challenge. Attention U-Net, DeepLabV3+ with an ImageNet pretrained encoder, and R2U-Net consistently produce good results in this task, with the latter being the only one to successfully extract the river in Figure 9a. Under certain conditions, nonwater features, such as fields and buildings, may produce reflectance characteristics similar to those of water in both the NIR and SWIR channels. Conversely, the tendency for water bodies to appear as low-intensity regions in the NIR and SWIR channels may be reversed, appearing brighter compared to the background. Such anomalies often result in either false positives or false negatives, respectively. Our findings show that many models are still capable of successful extraction under such conditions, suggesting that they can learn complex feature representations to identify water bodies. In particular, we observe that R2U-Net, DeepLabV3+, Attention U-Net, and U-Net++ produce fewer false positives, as seen in Figure 9b. All models demonstrated good performance on large water bodies, as can be seen in Figure 9c,d.

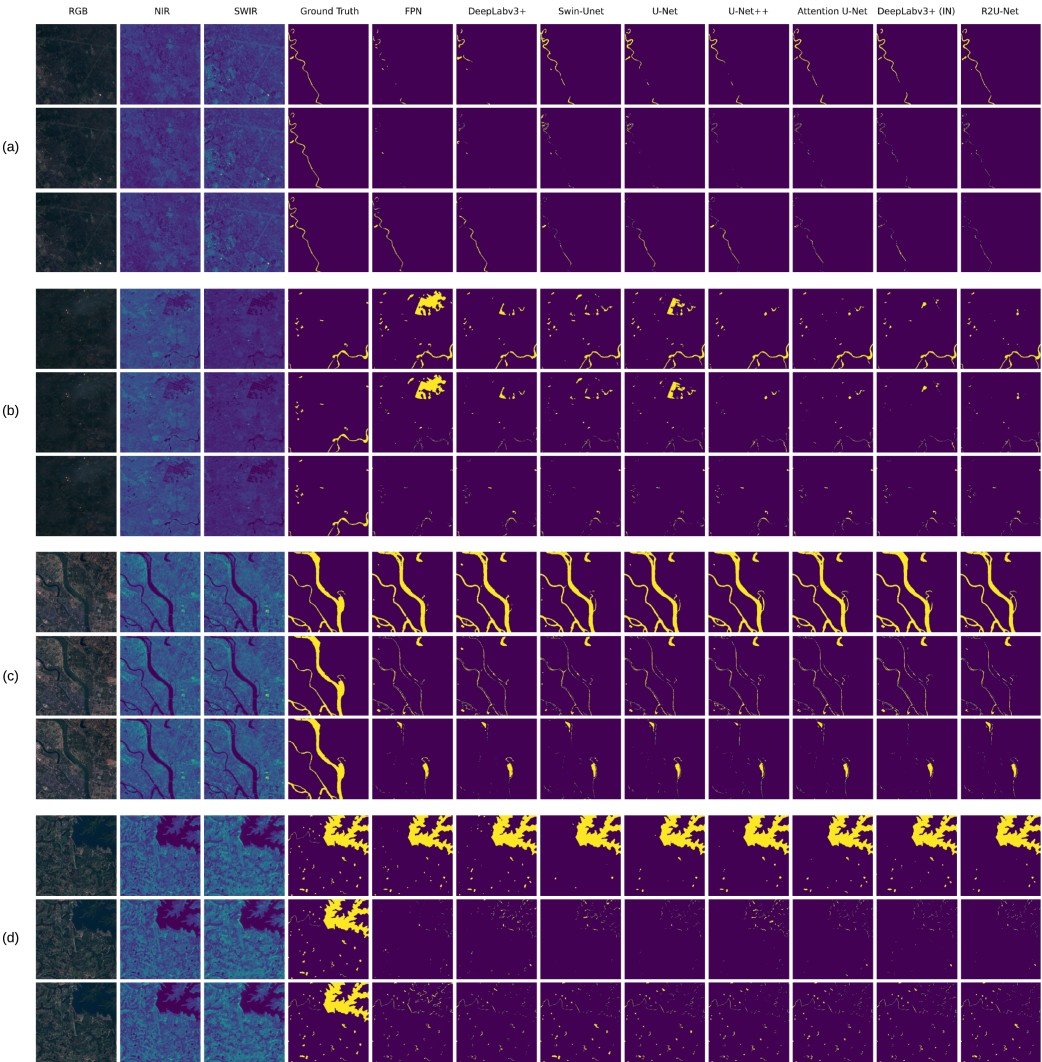

**Figure 9.** Qualitative comparison of model architectures on multispectral inputs. For each example, the first row shows raw predictions, the second row shows false positives, and the third row shows false negatives. (**a**) A thin river is only extracted fully by R2U-Net. (**b**) The fields in the top right produce false positives in many models. (**c**) A wide river is successfully extracted by all models. (**d**) A large contiguous water body is fully extracted by all models.

### 3.3. Various Spectral Band Contribution in Water Body Detection

In addition to comparing performances across models, Table 2 also provides a comparison between RGB, NIR, and multispectral inputs. It is unsurprising that multispectral models, having the most information with which to work, achieve the best results on our test data. We observed significant gains when going from RGB to NIR (2.764 mIoU) and from RGB to multispectral (4.193 mIoU). In contrast, there is a comparatively small difference between NIR and multispectral models, with an average gain of only 1.428 mIoU. These results suggest that the majority of relevant information is either introduced by or replicated in the NIR channel, with relatively little gained with the addition of RGB and SWIR. Figure 10 provides a qualitative comparison between input bands for FPN, U-Net, and R2U-Net. As suggested by our quantitative results, multispectral and NIR generally perform better than RGB across all models. This is especially true for thin rivers and micro-water bodies, as seen in Figures 10a,b. In contrast, we find there to be little difference between multispectral and NIR-only models. Our analysis shows that large contiguous water bodies, as seen in Figure 10d, are fully extracted by all models on each of the evaluated input bands.

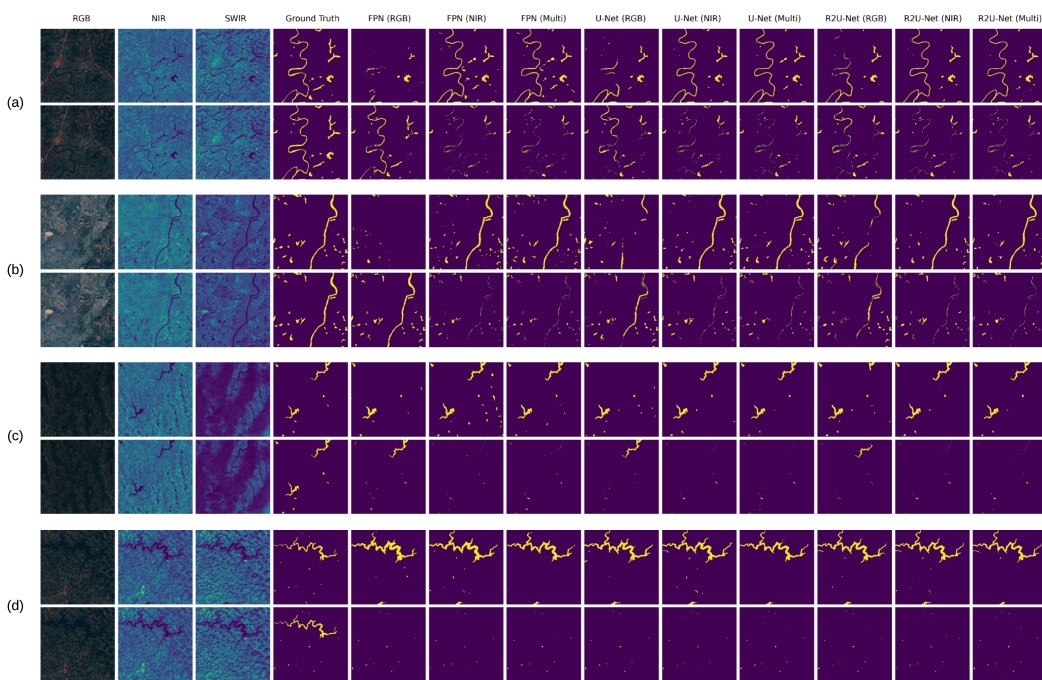

**Figure 10.** Qualitative comparison of RGB, NIR, and multispectral inputs for FPN, U-Net, and R2U-Net. For each example, the first row shows raw predictions and the second row shows false negatives. (**a**) A narrow, winding river is extracted by both NIR and multispectral models. (**b**) A river is extracted by both NIR and multispectral models, whereas RGB models struggle. (**c**) The lake in the upper right is fully extracted by only NIR and multispectral models. (**d**) A large contiguous water body is successfully extracted under all input bands.

### 3.4. Pixelwise Category Transplantation

Table 3 summarizes the performance of several multispectral models trained on data augmented by our proposed PCT algorithm. We previously defined $\theta$ as the minimum water content threshold at which we stop transplanting water bodies, which is used here as a hyperparameter set to values of 5%, 10%, 15%, and 20%. It is important to note that although $\theta$ correlates with the number of transplanted water bodies, there is no fixed quantity. Rather, we repeatedly transplant randomly selected water features until $\theta$ is exceeded. All models were trained exclusively on timestamp 1 (T = 1), with the exception of R2U-Net, which was also evaluated for timestamps 2 and 3 to demonstrate the robustness of our approach to a variety of atmospheric and lighting conditions. For each model, we report the baseline performance achieved without PCT, the performance after training with PCT, and the value for $\theta$ which produced the highest mIoU. The results in Table 3 demonstrate a consistent improvement across all models following the application of PCT. The most significant increase occurred in DeepLabV3+, with a gain of 1.302 mIoU, whereas the average gain from PCT was observed to be 0.749 mIoU. Results for R2U-Net on timestamps 1, 2, and 3 demonstrate that PCT can have a beneficial effect under a variety of conditions, including high cloud cover or low lighting. Figure 11 provides a qualitative comparison between Attention U-Net, DeepLabv3+, and R2U-Net before and after applying PCT. In particular, we observe improved extraction in all cases where the model was trained on PCT-augmented data.

**Table 3.** Performance after PCT.

| Model | Baseline (mIoU) | PCT (mIoU) | Best θ (%) | Gain (mIoU) |
|---|---|---|---|---|
| FPN [18] | 71.408 | 72.130 | 5% | 0.722 |
| DeepLabV3+ [19] | 72.032 | 73.334 | 15% | 1.302 |
| Swin-Unet [16] | 73.264 | 74.267 | 5% | 1.003 |
| Attention U-Net [17] | 74.763 | 75.269 | 5% | 0.506 |
| DeepLabV3+ (ImageNet) [19] | 75.403 | 75.728 | 15% | 0.325 |
| R2U-Net [15] ($T = 3$) | 74.828 | 75.657 | 10% | 0.829 |
| U-Net++ [14] | 74.551 | 75.445 | 5% | 0.894 |
| U-Net [13] | 74.532 | 75.546 | 10% | 1.014 |
| R2U-Net [15] ($T = 2$) | 75.424 | 75.710 | 10% | 0.286 |
| R2U-Net [15] ($T = 1$) | 75.420 | 76.024 | 15% | 0.604 |

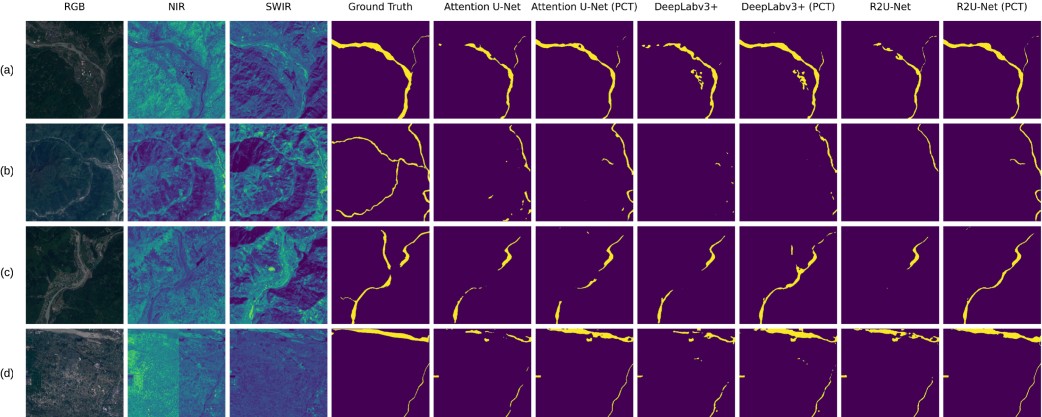

**Figure 11.** Qualitative comparison of Attention U-Net, DeepLabv3+, and R2U-Net before and after applying PCT. (**a**) Following PCT, all three models can extract the river along its full length. (**b**) A thin water body running through a mountainous area challenges all models, but improvement is observed following PCT. (**c**) All models initially struggle to extract a river running through a valley but demonstrate a substantial improvement after training with PCT. (**d**) Before PCT, all models struggle to fully extract the river running along the top of the sample. However, their performance on this task improves substantially after applying PCT.

## 4. Discussion

Motivated by the recent success of deep learning in water body extraction, we evaluated various state-of-the-art FCNs in this task. In particular, we find that U-Net-inspired architectures, which are characterized by symmetrically arranged encoder and decoder blocks, produce the best results. In contrast to index-based methods, we show that FCNs can successfully detect water under diverse conditions without the need for hand-tuned parameters. Moreover, they continue to perform well even in the presence of abnormal spectral signatures. This demonstrates that FCNs are able to learn rich feature representations, enabling them to recognize water bodies by their unique structural characteristics as opposed to relying on spectral signatures alone. Thus, they can perform well in situations in which index-based methods would typically fail. Furthermore, although traditional approaches depend on multiple spectral bands to detect water reliably, we observed relatively little difference between FCNs given NIR-only features versus those provided with the full multispectral dataset. This implies that they do not require the inclusion of multiple bands in order to suppress false positives arising from low-albedo surfaces. However, although we found that FCNs were able to produce consistently good results on large and medium-sized water bodies, the extraction of small and thin water bodies remains a challenging open research problem.

## 5. Conclusions

In this study, we evaluated the performance of several state-of-the-art FCNs on the task of water body extraction. Moreover, to address the shortcomings of previous works, we assessed each model on RGB, NIR, and multispectral features. Our results showed that R2U-Net [15] trained with a combination of the BCE and Jaccard losses achieved the best performance of 75.420 mIoU. Unsurprisingly, models trained on multispectral features produced the best results overall, outperforming RGB-only methods by an average of 4.193 mIoU. However, we observed comparatively little difference between multispectral and NIR-only methods, with a gain of only 1.428 mIoU on average. To address the issues of small datasets and class imbalance, we proposed our PCT algorithm as a novel approach to data augmentation. Experimental results demonstrated an average gain of 0.749 mIoU, with improvement observed across all evaluated models and datasets. Following the application of PCT, we were able to achieve a performance of 76.024 mIoU with multispectral R2U-Net, which outperforms the previous high of 75.130 mIoU attained by MC-WBDN [10] on the same dataset. Given these results, we believe that our approach may be applicable to a variety of problems in the domain of supervised image segmentation beyond the scope of water body extraction.

**Author Contributions:** Conceptualization, I.C.; methodology, J.B., S.I. and X.S.; software, J.B.; validation, J.B.; formal analysis, J.B.; investigation, J.B.; resources, I.C.; data curation, J.B.; writing—original draft preparation, J.B.; writing—review and editing, I.C., M.S.I. and X.S.; visualization, J.B.; supervision, I.C. and M.S.I.; project administration, I.C.; funding acquisition, I.C. All authors have read and agreed to the published version of the manuscript.

**Funding:** This research was funded by Natural Sciences and Engineering Research Council of Canada (NSERC) grant numbers 540034 and CRDPJ 543428-19.

**Data Availability Statement:** The initial dataset was collected by Yuan et al. [10] and is available at https://github.com/SCoulY/Sentinel-2-Water-Segmentation (accessed on 12 August 2021). Our code and PCT augmented dataset are available at https://github.com/JoshuaBillson/Waterbody-Detection-Via-Deep-Learning (accessed on 19 August 2022).

**Conflicts of Interest:** The authors declare no conflict of interest. The funders had no role in the design of the study; in the collection, analyses, or interpretation of data; in the writing of the manuscript; or in the decision to publish the results.

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
