# Peer review of "Water Body Extraction from Sentinel-2 Imagery with Deep Convolutional Networks and Pixelwise Category Transplantation"

_remotesensing, doi:10.3390/rs15051253_

Round 1
Reviewer 1 Report
The effort water body extraction using remote sensing data and DCNN is interesting and will bring a significant contribution in this field.
Besides this manuscript needs some improvement.
It is a good idea to add some description of the research area, maybe add some map with its location.
L135. Reference should be added to demonstrate “previously demonstrated promising results.”
It is a good idea to describe hardware that were used in this study. Also would be good to add short description of software that were used. Was is TF, Torch or anything else?
Section 2 Materials and methods should be better cited in my opinion. For example, L174 with BCE
I wish that my comment would be helpful in improving the quality of this research.
Thank you.
Reviewer 2 Report
In this study, authors have compared various FCN architectures for the extraction of water bodies from sentinel-2 imagery. Authors have also compared various loss functions and analyzed their performance for water body extraction. They also presented a new data augmentation technique named PCT which is helpful for dealing with the problem of class imbalance.
· Page number 3, line number 117-119 rewrite the sentence
· Page number 3, line number 97-98, “relatively little attention has been paid to the impact of band selection”. What do you mean by this? Whether earlier band selection was considered or not that is not clear.
· It is already proven in the literature that NIR and SWIR bands are useful for water & vegetation extraction applications, then what is the contribution of this study?
· Apart from the introduction of a new data augmentation technique named PCT, novelty is not clear in the study.
· Page number 15, conclusions section, line number 394, “we observed the best performance among multispectral models”. It is already proven in the literature that multispectral bands are helpful in the extraction of various features in remote-sensing images.
· Page number 15, conclusions section, line number 398-399, an average gain of 0.857 mIoU was observed when applying PCT for water body extraction. It is not a significant improvement.
Reviewer 3 Report
Based on Deep Convolutional Networks and Pixelwise Category Transplantation, the author has unveiled a detailed in this study. The topic is interesting for the journal's readership. Also, the methods are right and enough and the results are very good and reasonable. However, in my opinion, some issues should be addressed and improved before the manuscript can be published.
In the introduction section; please show the possible drawbacks of why you want to do this study.
Study Area: please give the time period for the population, temperature, precipitation, and other parameters.
Results of driving forces on surface water field variations: some sentences are the results, not the discussion. Could you please check all parts of the discussion?
Generally, the discussion is based on the main results or possible problems.
It is better to discuss the uncertainty in the discussion section.
Modify the conclusion section as per the study.
Reviewer 4 Report
The authors proposed Pixelwise Category Transplantation augmentation for Water Body Extraction from Sentinel-2 Imagery and compared various Deep Convolutional Neural Networks.
However, they should improve the introduction section and the accuracies of the previous studies should be added as well as a comparison between the tested losses functions in various applications. Moreover, the system requirements and algorithms training times should be addressed. Also, a comparison between the proposed approach and the traditional methods should be performed to demonstrate its improvements.
Additionally, the testing data which used in this study for comparison are very small compared to the training data only 65 images from 400. I suggest split the data with 250 training, 50 validation, and 100 testing. Furthermore, some figures are not clear and should be modified so that the final publication has a good impact.
In general, there structure and many English language errors scattered throughout the manuscript, for which the authors need to account for. The manuscript should have English proofreading.
Considering that the topic of the study is interesting, I suggest a reconsidering this manuscript after major review.
Please find below my comments and recommendations.
1- Introduction:
Lines 29-32: Please rephrase this paragraph.
Line 36: Please change the sentence (In (6) Feyisa et al.) to (Feyisa et al. (6)) and the same through the manuscript.
Lines 27-38: Please add more traditional studies for water body extraction with their achieved accuracies.
Line 74: Please add (Moreover,) before A modified U-Net.
Line 76: Please change the repeated (In ( ) ) word style.
Lines 79-82: Please rephrase this paragraph.
Lines 72-87: Please add the achieved accuracies to these studies.
Lines 91-92: Please rephrase this paragraph.
2- Materials and Methods.
Line 115: Please add the source of the ground truth data (how it was prepared).
Lines 132-136: This paragraph should be moved to the end of the introduction section.
Line 138: Please change the (tiles) word to images.
Line 142: Please change the (depicted by (a)) word (see figure 3-a) and the same through the manuscript.
Line 143: Please add (Moreover, or On the other hand) at the beginning of this sentence.
Line 153: Please remove the repeated illustration sentences here.
Line: Please add a reference to prove this hypothesis.
(It is a common practice to combine region-based losses such as Dice, Jaccard, and
Tversky with a pixel-wise loss such as BCE).
Lines 188-189: The testing data are very small compared to the training data only 65 images from 400. I suggest split the data with 250 training, 50 validation, and 100 testing.
Line 198: Please add a reference after (Adam algorithm).
Line 213-214: Please clarify this sentence and what is the meaning of (Hadamard)?
Line 215-225: The difference between the source and destination images is not clear the water body is the same only the background pixels changed I think this could increase over fitting possibility and reduce the generalization of the proposed approach. To evaluate the proposed approach, it should be tested using different datasets with various water body shapes.
3- Results.
Line 265: Please change Figure 6 to line graph and increase the colors contrast.
266: Please clarify Figure 7 as the differences between the resulted images is not clear Line. I suggest adding error subtraction images between the results and the ground truth image.
Line 278: Do you mean features?
295: Please clarify Figure 8 as the differences between the resulted images is not clear Line. I suggest adding error subtraction images between the results and the ground truth image.
Line 323: Please clarify how many images were transplanted in each case.
4- Discussion:
Lines 356-357: Please rephrase this sentence.
5- Conclusions:
Line 396: Please add mIoU after (only 1.381).
References:
All the references should follow the Remote Sensing journal style.
Round 2
Reviewer 2 Report
All the suggestions have been considered by the authors.
Reviewer 4 Report
The authors proposed Pixelwise Category Transplantation augmentation for Water Body Extraction from Sentinel-2 Imagery and compared various Deep Convolutional Neural Networks.
However, the authors performed some comments and the major comments still not revised. Moreover, some points need to be revised before the final approval as follows:
1- Again, there are structure and many English language errors scattered throughout the manuscript, for which the authors need to account for. Thus, the manuscript should have English proofreading.
2- Lines 23-37: The accuracy of the traditional index-based methods should be demonstrated in the introduction section.
3- Lines 72-88: Please remove the repeated words e.g. (achieving an F1 score of ….)
4- Lines 80-83: Please add a reference to prove this hypothesis.
5- Lines 72-88: Please discuss what is the improvement of the proposed method compared to the previous results considering they already achieved higher accuracies.
6- Line 346: Table1 (and 2): Please discuss why the achieved accuracies increased although the test data proportion increased. The resulted accuracy often has inversely relation with the test data proportion.
7- Line 369: Figures 8-9-10 Please rearrange the images so that each image followed vertically with the errors image.
8- The discussion section needs to be revised.
Thus, I suggest reconsidering this manuscript after performing the abovementioned comments.
